# ABCG2 transports anticancer drugs via a closed-to-open switch

Benjamin J. Orlando[1] & Maofu Liao [1✉]

ABCG2 is an ABC transporter that extrudes a variety of compounds from cells, and presents an obstacle in treating chemotherapy-resistant cancers. Despite recent structural insights, no anticancer drug bound to ABCG2 has been resolved, and the mechanisms of multidrug transport remain obscure. Such a gap of knowledge limits the development of novel compounds that block or evade this critical molecular pump. Here we present single-particle cryo-EM studies of ABCG2 in the apo state, and bound to the three structurally distinct chemotherapeutics. Without the binding of conformation-selective antibody fragments or inhibitors, the resting ABCG2 adopts a closed conformation. Our cryo-EM, biochemical, and functional analyses reveal the binding mode of three chemotherapeutic compounds, demonstrate how these molecules open the closed conformation of the transporter, and establish that imatinib is particularly effective in stabilizing the inward facing conformation of ABCG2. Together these studies reveal the previously unrecognized conformational cycle of ABCG2.

[1] Department of Cell Biology, Harvard Medical School, Boston, MA 02115, USA. ✉email: maofu_liao@hms.harvard.edu

The development of a multidrug-resistant (MDR) phenotype presents a significant challenge in the treatment of cancer with chemotherapy[1]. A well-established cause of MDR involves ATP-binding cassette (ABC) transporters, which are overexpressed in cancer cells and actively extrude anticancer drugs[2]. The most prominent MDR-related transporters, including P-glycoprotein (Pgp), ABCC1, and ABCG2, also critically affect pharmacokinetics, and regulate small molecule absorption in the intestine and transport across the blood–brain barrier[2,3]. Mechanistic understanding of the function and modulation of these transporters is of paramount importance in both cancer therapy and general pharmaceutical development.

Among the major human multidrug transporters, ABCG2 is unique in both structure and function. There are more than 200 known ABCG2 substrates which have highly diverse chemical structures and functions, and the substrate spectrum of ABCG2 is substantially overlapping with and yet distinct from that of Pgp or ABCC1[4]. ABCG2 transports uric acid and plays an essential role in the pathology of gout, but also confers resistance to a variety of chemotherapeutics including topoisomerase inhibitors (i.e., mitoxantrone (MXN), etoposide, topotecan, SN38), anthracyclines (i.e., doxorubicin and daunorubicin), antimetabolites (i.e., methotrexate and 5-fluorouracil), and photosensitizers (i.e., pheophorbide A)[5]. Transport of these compounds by ABCG2 is a limiting factor in their accumulation in cancerous cells, diffusion across the blood–brain barrier, and absorption through the intestine upon ingestion[6–9]. Potent inhibitors of ABCG2 have been identified and developed, with fumitremorgin C and the associated synthetic derivative Ko143 being the most commonly utilized compounds in vitro[10]. Despite the identification of a variety of compounds that block ABCG2, no specific ABCG2 inhibitor has yet attained clinical use[2,11]. Many efforts have been made to probe the binding sites and the interactions between diverse substrates and inhibitors[12–14], and the effects of site-specific mutations on the transport of different substrates[13,15–21]. These studies have suggested the existence of multiple substrate entry pathways and binding sites, which have yet to be revealed in high-resolution structural analyses.

Recent structural studies have revealed two conformational states of ABCG2: inward-facing conformation, in which the physiological substrate estrone-3-sulfate ($E_1S$) and various inhibitors bind in similar sites at the dimer interface of the transmembrane (TM) helices, and outward-open conformation, in which ATP binding induces nucleotide binding domain (NBD) dimerization and collapse of the substrate binding site[22–24]. Previous structures of ABCG2 in the inward facing conformation have been determined with the aid of a conformation specific 5D3 antibody fragment or high-affinity inhibitors, both of which are known to bias the conformational landscape of the transporter and potentially hinder analysis of important functional states in the transport cycle[25]. For this reason, it remains unclear whether additional transporter conformations are involved in polydrug recognition and transport. Importantly, no anticancer drugs have been resolved bound to ABCG2. Hence, the structural basis of polyspecific drug transport by ABCG2 remains poorly understood.

In this work, we used cryo-EM and biochemical assays to characterize the structures of ABCG2 in the unliganded state, and in the presence of the chemotherapeutic agents imatinib, MXN, or SN38. Our results show that in the absence of conformation specific antibody fragments or added ligands the transporter adopts a closed conformation, with substantial structural rearrangement of two critical transmembrane helices. Binding of chemotherapy drugs induced a conformational switch of the transporter, with different effects being observed when comparing imatinib to MXN or SN38. In total, our results delineate a new

paradigm for the conformational transition cycle of ABCG2, and reveal how different chemotherapy compounds distinctively alter the conformational space of the transporter.

## Results

**ABCG2 adopts a closed conformation in the resting state.** Since most ABCG2 structures determined to date have utilized conformation specific antibody fragments or high-affinity inhibitors, the conformation of ligand-free ABCG2 remains unclear. We subjected purified and nanodisc reconstituted human ABCG2 (Supplementary Fig. 1A–C) to single-particle cryo-EM analysis in the absence of antibody fragments or bound ligands (Supplementary Figs. 2 and 3). In the apo state, ABCG2 adopted a single predominant conformation that was distinct from that seen previously with the Fab or ligand bound transporter (Fig. 1a–c). In the absence of 5D3 fab or bound ligands the TM helices adopt a closed conformation while the NBDs remain separate in a nucleotide-free state (Fig. 1a). Thus, in our apo-closed conformation the arrangement of TM helices more closely resembles that seen in the outward facing ATP bound state (PDBid 6HBU)[24], whereas the lack of NBD dimerization more closely resembles that of the inward facing state (PDBid 5NJ3)[22].

In the apo-closed conformation the TM helices form a tightly packed helical bundle near the cytosolic region, primarily as a result of the intracellular region of TM5 residing closer to TM1 and TM2 of the opposite ABCG2 monomer (Fig. 1a–c). In the apo-closed conformation the position of TM5 seals the crevice between ABCG2 monomers where substrates and inhibitors bound in the previous inward facing structures[23,24]. The lateral shift of TM5 is reminiscent of the conformation previously observed in the ATP bound state of ABCG2[24]. However, in our apo-closed conformation TM5 is also rotated nearly 180° along the helical axis, thus generating an entirely different overall protein conformation (Fig. 1c, d). In previously determined inward facing ABCG2 structures the Phe545 phenyl moiety is positioned to the side of the central binding pocket and faces the same ABCG2 monomer to which it belongs[23,24]. In this location, Phe545 is buried in a pocket surrounded by the sulfur containing side-chains of Cys438, Met523, Met541, Cys544, and Met548 (Fig. 1d). However, the helical rotation of TM5 observed in our apo-closed structure causes the Phe545 sidechain to face the opposite ABCG2 monomer (Fig. 1b, c). Just above and below Phe545 along TM5 are four sulfur containing sidechains of Met541, Cys544, Met548, and Met549 (Fig. 1c). In the apo-closed conformation the positioning of TM5 results in these sidechains being oriented towards the dimer interface, such that a sulfur cluster of 10 cysteines and methionines along TM5 and TM2 is formed at the central 2-fold axis between ABCG2 monomers (Fig. 1c).

In addition to the translation and rotation of TM5, in the apo-closed conformation TM2 also displays an altered configuration compared to previous ABCG2 structures. Whereas the middle portion of TM2 (residues 434–438) adopts a helical character in the previous inward facing and ATP bound structures[23,24], this stretch of helix unwinds and bows outwards towards the lateral sides of the transporter in our apo-closed structure (Fig. 1b). The helical unraveling of TM2 causes structural rearrangement of several key residues previously identified in inhibitor and substrate bound ABCG2 structures. Phe439 has been shown to sandwich bound inhibitors and substrates through aromatic interactions in the inward facing conformation[23,24]. However, as a result of the helical unraveling of TM2 observed in our apo-closed structure the Phe439 phenyl moiety is flipped ~180° outward and away from the 2-fold axis (Fig. 1b, c). In this flipped state the Phe439 sidechain is positioned between TM1 and TM2

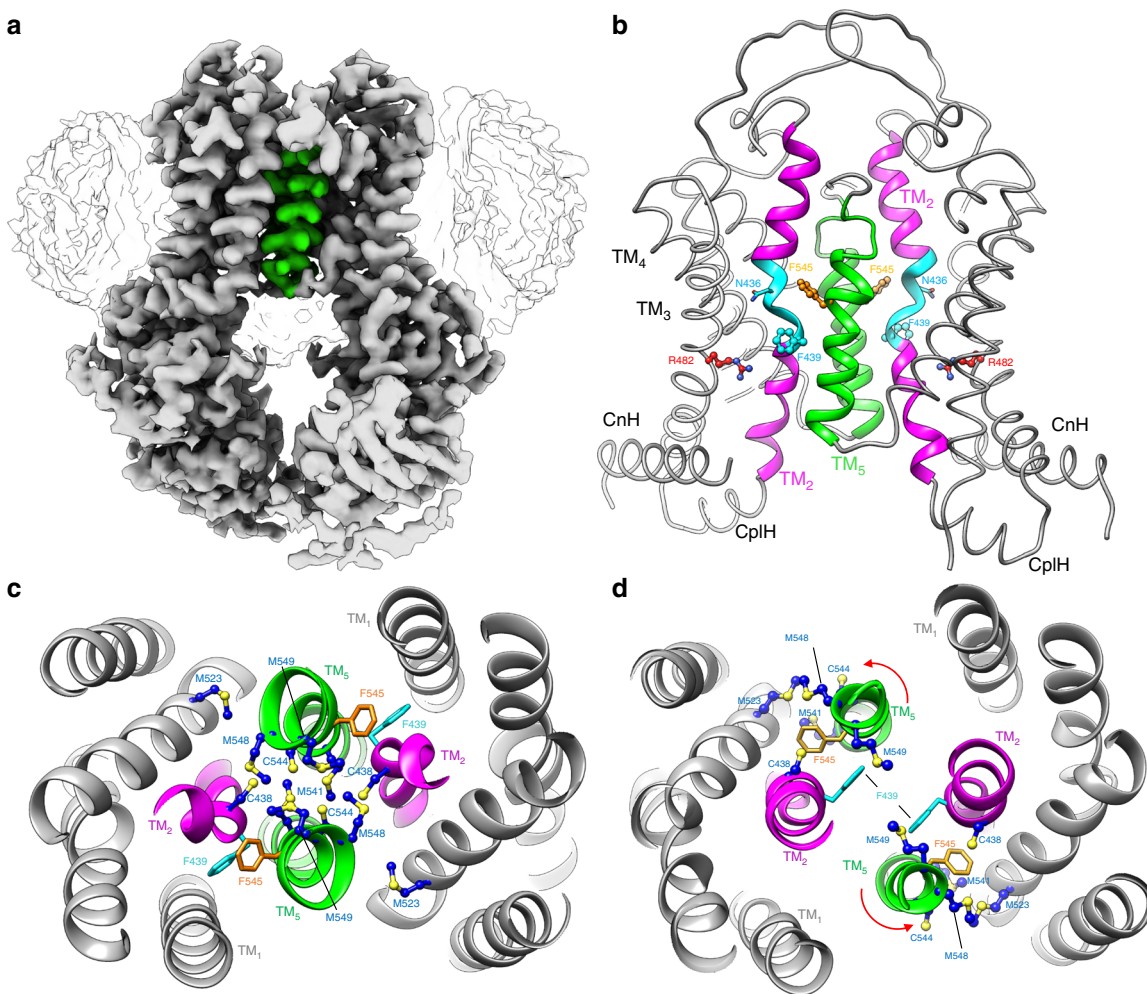

**Fig. 1 Cryo-EM structure of apo-closed ABCG2. a** Final cryo-EM map of nanodisc reconstituted ABCG2 in the absence of added ligands. Nanodisc density is shown in white, and TM5 is colored green. In the apo-closed conformation the TM helices are closed, while the NBDs remain open in a nucleotide-free state. **b** Overall arrangement of the TM helices in the apo-closed conformation. TM2 is colored magenta, with the central portion that unravels colored cyan. TM5 is colored green. **c** View from the extracellular space of the TM helix arrangement in the apo-closed conformation. In the apo-closed conformation, Phe439 (cyan sticks) is flipped outward away from the center of the transporter, and rotation of TM5 causes Phe545 (orange sticks) to point towards the opposite ABCG2 monomer, and brings several sulfur containing sidechains (blue and yellow sticks) to the central dimer axis. **d** View from the extracellular space of the TM helix arrangement in previously determined inward facing ABCG2 cryo-EM structures. TM helices and residues are colored the same as in (**c**). Red arrows indicate the direction of TM5 rotation in going from inward facing to apo-closed states.

of the same monomer, bringing it into closer proximity to Arg482 (Fig. 1b, c), which is a known hotspot for mutations resulting in altered transport substrate specificity[26,27]. Similarly, Asn436 in TM2 which forms hydrophilic interactions with bound inhibitors and transport substrates in the previous structures[23,24] is flipped outward and away from the 2-fold axis in the apo-closed conformation (Fig. 1b). In this conformation Asn436 also points in the general direction of the side pocket containing Arg482. Thus, key residues that have previously been identified as determinants of ligand binding and specificity, adopt drastically different conformations in the absence and presence of small molecule ligands or conformation specific antibody fragments.

**The apo-closed conformation of ABCG2 is sampled in vivo.** Observation of a closed TM helical bundle in the absence of added substrate, inhibitor, or NBD dimerization was highly unanticipated. In order to verify that the apo-closed conformation was not an artifact of purification or nanodisc incorporation,

we performed disulfide cross-linking analysis in intact cells. Wild-type ABCG2 contains a native disulfide bond between Cys603 of opposing monomers, which causes the protein to run at a dimeric molecular weight in non-reducing sodium dodecyl sulfate-polyacrylamide gel electrophoresis (SDS-PAGE) (WT: Fig. 2b, e). While mutation of Cys603 abolishes inter-monomer disulfide formation (ΔC: Fig. 2b, e), the mutation has previously been shown not to impact membrane targeting or transport function[28]. The translation and rotation of TM5 observed between apo-closed, inward, and nucleotide-bound states of ABCG2 provides a platform upon which to introduce cysteine mutations to monitor the conformation of ABCG2 in intact cells. We introduced two mutations in TM5 (Val534Cys and Ala537Cys) independently into a Cys603Ser background to monitor oxidant-induced inter-monomer disulfide formation (Supplementary Table 2). Val534-Cys was chosen to form inter-monomer disulfides only when ABCG2 is in the ATP bound state (Fig. 2a), whereas Ala537Cys should form inter-monomer disulfides only when ABCG2 is in the apo-closed conformation (Fig. 2d). When ABCG2 is in the inward facing conformation these residues are too distant to

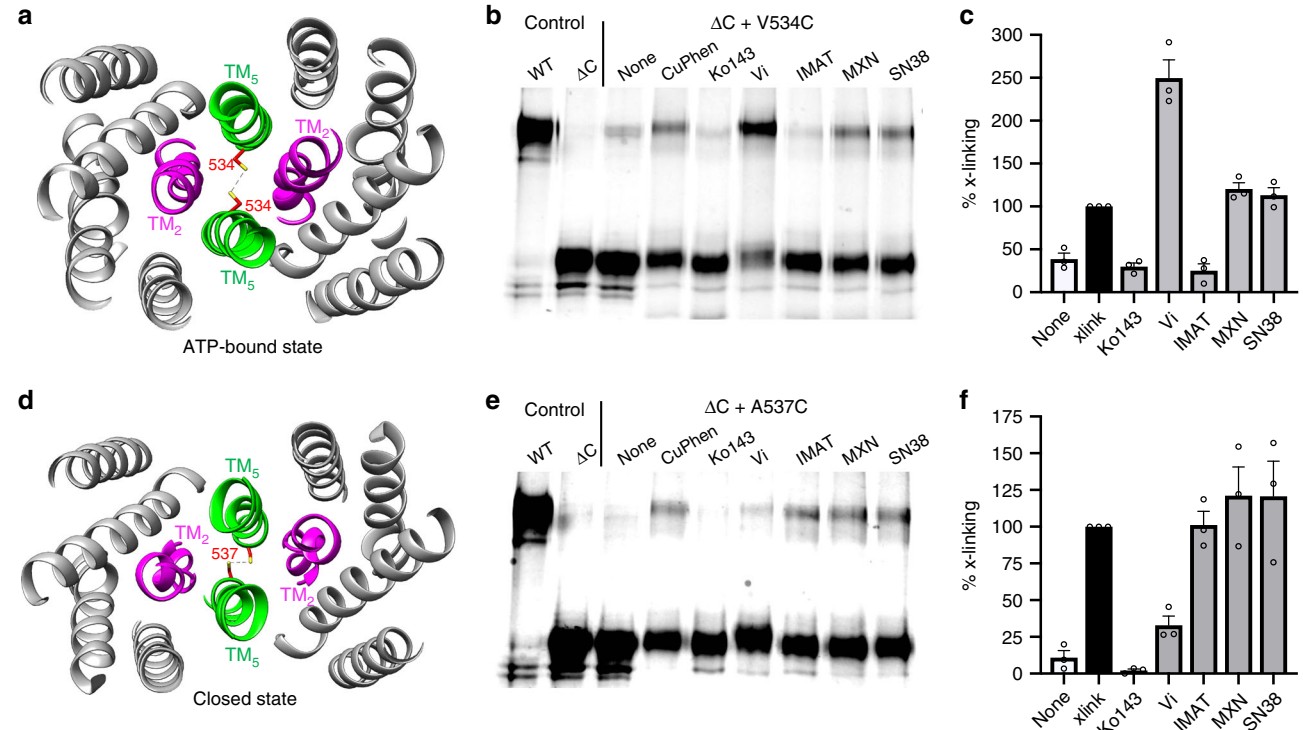

**Fig. 2 Disulfide crosslinking of ABCG2 in different conformational states. a** View from the extracellular space of ABCG2 TM helices in the ATP bound state (PDBid 6HZM). A cysteine was introduced at position 534 to probe disulfide crosslinking across the dimer interface. **b** Non-reducing SDS-PAGE with in-gel GFP fluorescence to monitor oxidant-induced crosslinking of the Val534Cys mutant. Upper band represents disulfide crosslinked ABCG2. **c** Densitometry quantification of crosslinking in (**b**). Data represents the average of three independent experiments with SEM. **d** View from the extracellular space of ABCG2 TM helices in the apo-closed state. A cysteine was introduced at position 537 to probe disulfide crosslinking across the dimer interface. **e** Non-reducing SDS-PAGE with in-gel GFP fluorescence to monitor oxidant-induced crosslinking of the Val534Cys mutant. Upper band represents disulfide crosslinked ABCG2. **f** Densitometry quantification of crosslinking in (**e**). Data represents the average of three independent experiments with SEM. Source data are provided as a Source data file.

form inter-monomer disulfides and no crosslinking should be observed.

The crosslinking mutants were expressed in HEK293F cells, which were subsequently treated with different transport inhibitors prior to oxidation with copper phenanthroline (CuPhen) to induce disulfide crosslinking. In the absence of added oxidant, the Ala537Cys mutant migrates primarily as a monomeric species indicating a lack of inter-monomer disulfide bonds, whereas a slight amount of crosslinked dimer species is present for the Val534Cys construct before the addition of oxidant. After the addition of CuPhen both mutants demonstrate the formation of a higher molecular weight species indicative of inter-monomer disulfide formation (Fig. 2b, e). The potent inhibitor Ko143 is known to induce an inward facing conformation of ABCG2, and application of this inhibitor prior to the addition of CuPhen prevented crosslinking in both mutants (Fig. 2b, e). Sodium orthovanadate (V$_i$) is a potent inhibitor of ABCG2 ATPase activity that locks the transporter in an ATP bound conformation by trapping a transition state intermediate during ATP hydrolysis. Application of V$_i$ to mutant expressing cells potentiated the degree of crosslinking observed with the Val534Cys mutant, indicating that V$_i$ efficiently locked ABCG2 into a nucleotide bound state (Fig. 2b). On the other hand, V$_i$ prevented crosslinking of the Ala537Cys mutant, consistent with the cryo-EM structures demonstrating that this mutant should only form disulfides when ABCG2 is in the apo-closed conformation and not the nucleotide bound state (Fig. 2e). Thus, our crosslinking analysis not only confirms that the Val534Cys

and Ala537Cys mutants serve as accurate reporters of the conformational state of ABCG2 in intact cells, but also that the apo-closed conformation observed in our cryo-EM structure occurs in vivo.

**Conformational dynamics of ABCG2 during multidrug transport.** Our structural and biochemical characterizations have suggested a general conformational transition cycle for ABCG2. However, it is unclear how the observed conformational dynamics are utilized to facilitate the transport of a wide variety of compounds. In order to understand how ABCG2 interacts with chemotherapy drugs of diverse chemical structure, we selected three representative chemotherapy drugs with highly divergent chemical scaffolds: imatinib, mitoxantrone (MXN), and SN38. Imatinib is a landmark drug that revolutionized the treatment of chronic myelogenous leukemia[29]. Whether imatinib is a substrate or inhibitor for ABCG2 is a subject of debate, due to various properties of the drug shown in different studies[30–32]. MXN and SN38 are topoisomerase inhibitors that have been extensively characterized as ABCG2 substrates[33,34]. In order to assess the effect of these three chemotherapy compounds on ABCG2 structure and function we performed a variety of biochemical assays with each compound.

We first compared the effect of each chemotherapy drug on ABCG2 conformation with the disulfide crosslinking assay described above. Compounds that stabilize the inward-facing conformation are expected to prevent crosslinking of both the

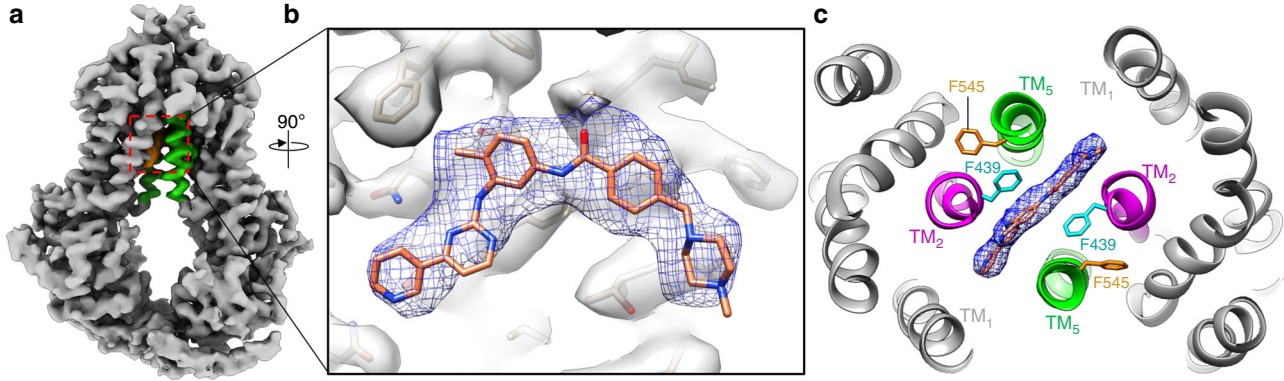

**Fig. 3 Imatinib traps ABCG2 in an inward conformation. a** Cryo-EM map of ABCG2 bound to imatinib. ABCG2 is in an inward facing conformation. TM5 is colored green, and imatinib is colored orange. Red dashed box indicates the position of bound imatinib. **b** Zoomed-in and rotated view of the highlighted area in (**a**) showing the cryo-EM map for imatinib. **c** View of the imatinib binding pocket from the extracellular space. Imatinib binds on the 2-fold axis between ABCG2 monomers and is sandwiched between Phe439 sidechains from each monomer.

Val534Cys and Ala537Cys mutants. Indeed, the potent transport inhibitor Ko143, which is known to stabilize the inward facing conformation, attenuated crosslinking of both mutants (Fig. 2b, e). Imatinib prevented crosslinking of the Val534Cys mutant, demonstrating that, similar to Ko143, this compound prevents ABCG2 from entering the nucleotide bound outward facing conformation (Fig. 2b, c). Consistent with this result we observed that imatinib inhibited the ATPase activity of nanodisc reconstituted ABCG2 to levels comparable to that seen with the potent inhibitor Ko143 (Supplementary Fig. 1D). In a 5D3 antibody shift assay[35] imatinib promoted antibody binding to the surface of intact cells similar to Ko143 (Supplementary Fig. 1F), further indicating that imatinib prevents ABCG2 from cycling through the ATP bound conformation. However, unlike Ko143, imatinib did not prevent crosslinking of the Ala537Cys mutant in intact cells (Fig. 2e, f), suggesting that although imatinib can prevent the formation of the nucleotide bound outward facing state, it is not as effective as Ko143 in stabilizing the inward facing conformation. Consistent with this notion we found that while Ko143 induced a significant thermostabilizing effect on ABCG2 (~3 °C), imatinib was less effective and only stabilized ABCG2 by ~1 °C (Supplementary Fig. 1E). Overall the results of our functional assays with imatinib suggest that the molecule can inhibit ATP hydrolysis by preventing the formation of a nucleotide bound outward facing conformation, and likely stabilizes the inward facing conformation of ABCG2 similar to the potent inhibitor Ko143.

In contrast to imatinib, the bona fide transport substrates MXN and SN38 had little to no effect on oxidative crosslinking of the Val534Cys or Ala537Cys mutants (Fig. 2b, e). The lack of an effect on crosslinking suggests that MXN and SN38 do not stabilize ABCG2 in any one particular state under normal cellular transport conditions. Similarly, MXN and SN38 showed no effect on the binding of 5D3 antibody to intact cells (Supplementary Fig. 1G, H), indicating that they did not stabilize the inward facing conformation like Ko143 or imatinib. Consistent with these results we observed that MXN and SN38 had little to no effect on the ATPase activity of nanodisc reconstituted ABCG2, similar to what was previously observed with the transport substrate $E_1S$ and ABCG2 in lipid nanodiscs[22] (Supplementary Fig. 1D). Similarly, MXN and SN38 did not display any significant effect on the thermostability of ABCG2 (less than 1 °C, Supplementary Fig. 1E). Together these results demonstrate that unlike molecules such as Ko143 or imatinib that effectively stabilize certain conformations of ABCG2, transport substrates

such as MXN and SN38 have less of an effect on the conformational landscape of the transporter.

In summary, our biochemical experiments with three different chemotherapeutics demonstrate differential effects on ABCG2 conformation and function between imatinib or the bona fide substrates MXN and SN38. To further dissect the impacts of these drugs on the conformational state of ABCG2, we pursued single-particle cryo-EM analysis of nanodisc reconstituted ABCG2 in the presence of each individual chemotherapeutic compound.

**Imatinib traps ABCG2 in an inward facing conformation**. The structure of ABCG2 bound to imatinib was determined in the same fashion as the apo-closed conformation, except that the nanodisc reconstituted transporter was incubated with imatinib on ice for ~45 min before plunge freezing cryo-EM grids. Three-dimensional (3D) classification of ABCG2 particles bound to imatinib demonstrated a single predominant conformation with the TM helices in an inward open facing conformation (Supplementary Fig. 4). The inward open conformation observed in the presence of imatinib is essentially identical to the conformation seen previously with ABCG2 bound to 5D3 Fab fragments or high affinity inhibitors[22–24]. Throughout 3D classification we observed no stable classes with closed TM helices corresponding to the apo-closed conformation. Thus, our 3D classification demonstrates that imatinib was effective in stabilizing ABCG2 in the inward facing conformation, consistent with the results of disulfide crosslinking (Fig. 2), 5D3 antibody shift (Supplementary Fig. 1E), and ATPase assays (Supplementary Fig. 1D).

The final cryo-EM structure of imatinib-bound ABCG2 at an overall resolution of 4.0 Å revealed one imatinib molecule bound in the inward-facing cavity between ABCG2 monomers (Fig. 3a–c). Imatinib bound in a location that spans across the top of the inward-facing cavity just beneath the leucine plug, similar to the previous binding position of the transport substrate $E_1S$ and the inhibitor MB136[23,24]. Imatinib is sandwiched via π-stacking interactions between the Phe439 phenyl moiety from opposing ABCG2 monomers (Fig. 3c). Although bound in a similar location as previously seen with the inhibitor MB136, imatinib occupies significantly less volume at the dimer interface (Supplementary Fig. 10). Moreover, imatinib is bound relatively high in the inward facing cavity, whereas inhibitors such as MB136 and MZ29 extend farther towards the cytosol and make additional contacts with TM1 (Supplementary Fig. 10). Although imatinib occupies a smaller volume and makes fewer protein contacts than other ABCG2 inhibitors, the molecule similarly acts

like a wedge to force ABCG2 into the inward facing conformation and ultimately prevent ATP hydrolysis. Taken together, the difference in binding details of imatinib and Ko143-derived inhibitors may be underlying the observed different biochemical properties for imatinib and Ko143 in our crosslinking and thermoshift assays.

**Structures of ABCG2 bound to MXN or SN38**. The structures of ABCG2 bound to MXN or SN38 were determined in the same fashion as the imatinib bound structure, by incubating the nanodisc reconstituted transporter with either compound on ice for ~45 min before plunge freezing cryo-EM grids. In contrast to the situation observed with imatinib, 3D classification of ABCG2 particles in the presence of MXN (Supplementary Fig. 6) or SN38 (Supplementary Fig. 8) consistently revealed two predominant conformations. Subsequent refinement of particles belonging to either conformation demonstrated that the two observed states correspond to the apo-closed and inward facing conformations. The refined apo-closed conformation in the presence of MXN or SN38 was identical to that observed in the absence of added transport substrate, and we observed no additional clear EM density that can be accounted for with bound drug molecules. Thus, in the presence of transport substrates MXN or SN38 a significant fraction of ABCG2 particles appear not to bind the substrates and remain in the apo-closed conformation. This is in contrast to the situation observed with imatinib, in which no subset of particle images corresponding to the apo-closed conformation can be obtained through classification. Thus, consistent with the variations observed in our biochemical experiments, our cryo-EM single-particle analysis clearly revealed distinct effects on the conformation of ABCG2 induced by imatinib and the bona fide substrates MXN or SN38.

Refinement of ABCG2 particles in the inward facing conformation produced final maps of the transporter bound to MXN (Supplementary Fig. 7) or SN38 (Supplementary Fig. 9) at 3.6 and 4.1 Å overall resolution, respectively. MXN and SN38 bound in the crevice between ABCG2 monomers, which largely overlaps with the binding sites of the physiological substrate $E_1S$ and small-molecule inhibitors, all of which are in the same inward-facing conformation[23,24]. An inherent complication in resolving drug-bound structures of ABCG2 with cryo-EM lies in the fact that the drug binding pocket lies directly along the 2-fold axis relating monomers of ABCG2. As a result, the particles in 3D reconstruction contain inherent 2-fold pseudo-symmetry, which causes the density for drug molecules to suffer from symmetry artifacts. Several attempts to break this pseudo-symmetry through symmetry expansion and/or classification without alignment were unsuccessful. Nevertheless, the drug densities for MXN, SN38, and imatinib are of sufficient quality to determine the overall binding orientation of each molecule. A consistent feature of ABCG2 binding seen with all substrates and inhibitors is that the aromatic rings of the bound compounds are sandwiched through π-stacking interactions with the phenyl moiety of Phe439 from opposing ABCG2 monomers (Fig. 4a–f). The anthracene rings of MXN point vertically along the 2-fold axis between ABCG2 monomers, and the molecule is oriented such that the hydroxyl-ethylamine tails are positioned more towards the cytoplasm (Fig. 4c). The anthracene rings of MXN and the phenyl moiety of the Phe439 sidechain are offset in a parallel displaced configuration (Fig. 4b). The top of the MXN anthracene ring interacts through a π-sulfur interaction with Met549, and the hydroxyl-ethylamine tails penetrate into a pocket between TM1 and TM2, forming a hydrogen bond with Asn436 (Fig. 4c). SN38 binds in the same general location and forms a hydrogen bond between the A-ring hydroxyl and Asn436 in addition to the π-stacking interaction with Phe439 (Fig. 4d–f). Overall, the binding position of MXN and SN38 in the inward-facing conformation is consistent with that seen previously for the substrate $E_1S$, suggesting that all transport substrates may eventually diffuse from the cytosol or the inner membrane leaflet into this same inward-facing pocket. The same ring-stacking interaction between the Phe439 sidechain and all bound compounds at least partially serves as the basis of the large spectrum of ABCG2 substrates, all of which contain aromatic ring structures.

## Discussion

In this work, we have shown that ABCG2 adopts a previously unrecognized apo-closed conformation in the absence of bound ligands or conformation specific antibody fragments. In the apo-closed state the NBDs remain well separated similar to previous inward facing ABCG2 structures, but structural rearrangement of the TM helices generates an entirely different protein conformation. In the apo-closed conformation TM5 is shifted laterally and rotated ~180° towards TM1 and TM2 of the opposite monomer, sealing off the crevice at the dimer interface where ligands and inhibitors have bound in the previous inward facing ABCG2 structures. Furthermore, in the apo-closed conformation the middle portion of TM2 unravels and loses helical character. This helical unraveling causes Phe439 and Asn436, residues known to be critical determinants of ligand binding[24], to be oriented outward away from the dimer axis. As a result of the altered TM2 and TM5 configurations, the previously identified ligand binding pocket between ABCG2 monomers is completely absent in the apo-closed conformation. Importantly, site-specific cysteine crosslinking confirmed that the apo-closed conformation is sampled in a cellular context (Fig. 2d, e), validating that this state of the transporter is a relevant intermediate in the conformational cycle of ABCG2 mediated drug transport.

Our biochemical and structural analyses demonstrate that different ligands can induce a conformational shift of ABCG2 from the apo-closed to inward facing conformation. Molecules such as imatinib and Ko143 inhibit ATPase activity, promote binding of the 5D3 antibody, and favor stabilization of the inward facing conformation. On the other hand, transport substrates such as MXN and SN38 are less effective in this regard, and displayed little effect on thermostability, ATPase activity, 5D3 binding, or disulfide crosslinking. Nevertheless, cryo-EM reconstructions of ABCG2 bound to MXN or SN38 revealed that both molecules bound in the crevice between ABCG2 monomers in the inward facing conformation, analogous to that seen with imatinib. However, in the presence of MXN and SN38 a significant proportion of particles remained in the apo-closed conformation, indicating that neither compound was as effective as imatinib in stabilizing the inward facing state.

Together our structural and functional data reveal a previously unrecognized conformational cycle of ABCG2 (Fig. 5). In this model, ABCG2 is first in the apo-closed conformation with the TM helices collapsed and the NBDs open (Fig. 5, state 1). Binding of transport substrates or inhibitors causes the protein to shift toward the inward facing conformation, where the small molecules are trapped primarily through hydrophobic interaction at the dimer interface (Fig. 5, state 2). In the case of transported substrates, subsequent ATP binding would promote the efflux of the substrate to the extracellular space (Fig. 5, state 3)[36], followed by ATP hydrolysis to reset the transporter to the apo-closed conformation. On the other hand, binding of transport inhibitors in the central cleft between monomers (Fig. 5, state 2) prevents ATP binding/hydrolysis and further conformational change to the outward facing state.

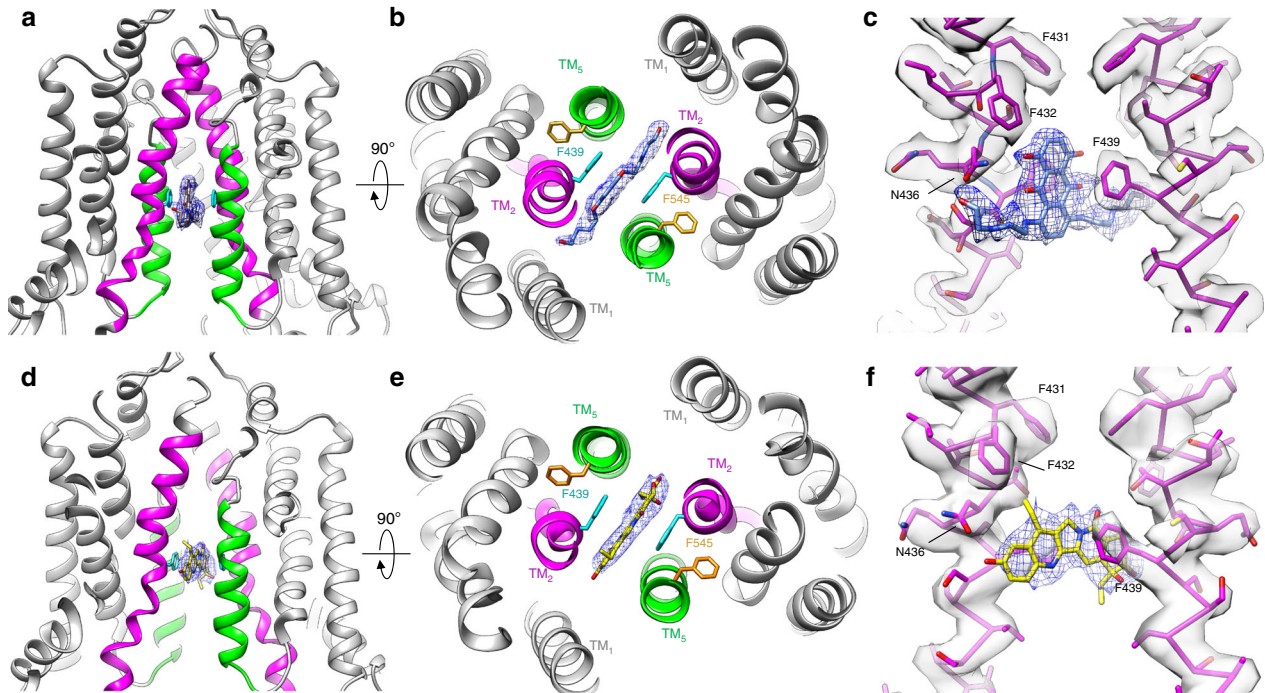

**Fig. 4 Binding of MXN and SN38 to ABCG2. a** View of ABCG2 TM helices in the inward conformation bound to MXN. TM2 is colored magenta, and TM5 is colored green. **b** View from the extracellular space showing that MXN binds between Phe439 from opposing monomers. **c** Zoomed-in view of the MXN binding site with MXN shown in blue. TM2 and associated density from each monomer is shown as magenta sticks. **d** View of ABCG2 TM helices in the inward conformation bound to SN38. TM2 is colored magenta, and TM5 is colored green. **e** View from the extracellular space showing that SN38 binds between Phe439 from opposing monomers. **f** Zoomed-in view of the SN38 binding site with SN38 shown as yellow sticks. TM2 is shown as magenta sticks.

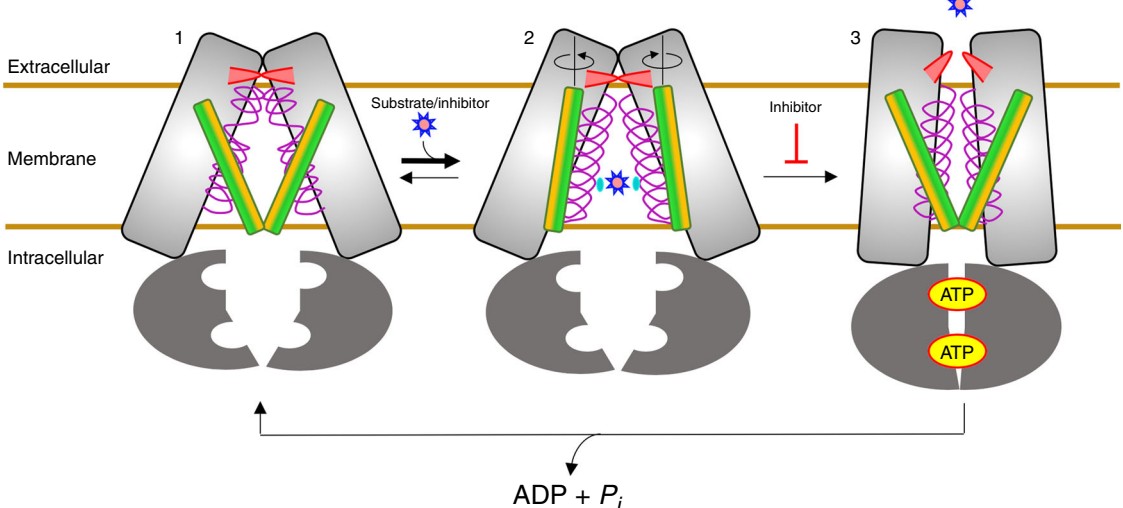

**Fig. 5 Model of drug capture and extrusion by ABCG2.** In the absence of ligands ABCG2 adopts an apo-closed conformation (1). In this conformation the NBDs are well separated, TM5 (green/yellow bar) is shifted towards the dimer axis, and TM2 (purple spring) is unwound near the center of the helix. Binding of substrates and inhibitors shifts ABCG2 to the inward facing conformation (2). In this conformation TM2 regains helical character, TM5 moves laterally to the sides of the transporter and rotates away from the dimer axis. Bound drug molecules are captured between Phe439 from each monomer (cyan circle). Subsequent binding of ATP in the NBDs causes the TM helix bundles to collapse toward one another, pushing the bound drug molecule through the leucine plug (red flipper) and into the extracellular space. ATP hydrolysis resets the transporter to the apo-closed conformation.

In the scheme proposed in Fig. 5, ABCG2 inhibitors and substrates would be proposed to interact directly with the closed conformation of the transporter to induce a conformational shift to the inward facing state. However, we cannot exclude the possibility that in the resting state, ABCG2 samples less stable conformations other than the most stable apo-closed conformation. It is possible that rather than binding directly to and altering the closed conformation, small molecules such as imatinib, MXN, and SN38 may stabilize the inward open state through a process of conformational selection. Indeed, through-out 3D classification of the apo ABCG2 dataset, we detected several 3D classes with well-defined nanodisc and NBD densities

despite a lack of stable homogenous TM helices, indicating that ABCG2 likely samples various conformational states despite the apo-closed conformation being the most stable and readily classified (Supplementary Fig. 2). Whether or not small molecules interact directly with the apo-closed conformation and induce a shift to the inward facing state, or stabilize the inward facing state through a process of conformational selection will require further investigation.

Consistent with previous studies, our cryo-EM structures demonstrate that both transported substrates and inhibitors bind in largely overlapping locations (Supplementary Fig. 10)[23,24]. When ABCG2 is in the inward facing conformation, imatinib, MXN, and SN38 all bound in the crevice between ABCG2 monomers where they are sandwiched between Phe439 sidechains (Figs. 3b and 4b, e and Supplementary Fig. 10). Although these molecules bind in largely overlapping sites, they display significantly different effects on the ATPase activity, 5D3 reactivity, and conformational state of ABCG2. Whereas imatinib acts as a wedge to prevent ATP-induced NBD dimerization and TM helix collapse, the substrates MXN and SN38 can move from their initial binding sites and be translocated. As has been suggested previously, a higher affinity and larger molecular volume of imatinib may explain why this molecule can act as a wedge and prevent the formation of a nucleotide locked state[23]. However, the fact that mutation of Arg482, a residue that is outside the interaction range of imatinib in our structure, can completely reverse the inhibitory and 5D3 reactivity effects seen with this molecule[26], supports the idea that binding affinity and/or molecular volume alone may not be sufficient to completely explain the observed inhibitory and conformational effects. The fact that Arg482 interacts with the region of TM2 that is unraveled in our apo-closed structure seems to suggest that modulating the dynamics of TM2 is a key factor in discrimination between transported substrates and inhibitors. Further structural and biochemical analyses will be required to reveal the intricacies of how some compounds behave as inhibitors while others are transported substrates, and how mutations outside of the identified drug binding pocket can allosterically alter these properties.

Due to a lack of domain swapping and extension of the TM helices into the cytosol, ABCG2 is structurally distinct from the other two major human multidrug transporters Pgp and ABCC1. As a result of this difference in architecture, the mechanisms of substrate capture and transport are also distinct when comparing ABCG2 vs Pgp and ABCC1. In the resting state Pgp and ABCC1 adopt rather open conformations in the TMDs and NBDs[37,38]. Pgp uses a large globular hydrophobic pocket with many aromatic side chains to capture generally hydrophobic molecules[37], whereas ABCC1 uses pockets in the TMDs to recognize a hydrophobic and hydrophilic portion of its preferred substrates such as leukotriene C4[38]. In both cases, specific substrate binding induces a closure of the TMDs around the substrate, and accordingly brings opposing NBDs into closer proximity to be primed for ATP binding and hydrolysis. In contrast, ABCG2 adopts a closed conformation in the resting state and specific substrate binding induces an opening of the TMDs. Thus, ABCG2 likely selects its substrates by sensing whether the compound can effectively shift the initial apo-closed conformation to an inward open state. Although the closed-to-open switch in ABCG2 appears opposite to the open-to-closed transition in Pgp and ABCC1, they are conceptually similar events in which substrate selection is tightly coupled with conformational changes of the transporters.

## Methods

**Expression and purification of human ABCG2.** Human ABCG2 with an N-terminal maltose binding protein fusion (MBP-ABCG2) was expressed in HEK293F (ThermoFisher—R79007) cells using BacMam virus transduction.

HEK293F cells were grown to a density of ~2E6 cells/mL before the addition of BacMam virus, and after ~12 h the culture was supplemented with 10 mM sodium butyrate and the temperature reduced from 37 to 30 °C. Protein expression was allowed to proceed for ~60–72 h before harvesting cells via centrifugation. Cell pellets were stored at −80 °C until the time of purification.

For purification of MBP-ABCG2 cell pellet corresponding to 2 L of culture volume was resuspended with a glass Dounce homogenizer in buffer containing 25 mM Tris (pH 8), 150 mM NaCl, and Roche cOmplete ethylenediaminetetraacetic acid (EDTA) free protease inhibitor cocktail. A mixture of dodecyl-β-D-maltopyranoside (DDM)/cholesteryl hemisuccinate (CHS) was added to a final concentration of 1% DDM and 0.2% CHS, and membranes were solubilized by stirring the mixture for 90 min at 4 °C. Large debris was removed by low speed centrifugation, and insoluble material was pelleted by ultracentrifugation at ~100,000 × g for 1 h at 4 °C. The resulting supernatant was filtered and applied to amylose affinity resin in a gravity flow format. The resin was washed with 10 column volumes of 25 mM Tris (pH 8), 150 mM NaCl, 0.05% DDM, 0.01% CHS before eluting the bound MBP-ABCG2 with the same buffer containing 10 mM maltose. Purified MBP-ABCG2 was concentrated in a 100 kDa molecular weight cut-off (MWCO) spin concentrator to ~5 mg/mL.

Concentrated MBP-ABCG2 was incorporated into lipid nanodiscs by mixing the purified protein and a MSP1D1 scaffold protein and a cholate solubilized mixture (w/w) of 80% POPC (1-palmitoyl-2-oleoyl-glycero-3-phosphocholine) and 20% POPS (1-palmitoyl-2-oleoyl-sn-glycero-3-phospho-L-serine) at a ratio of 1:20:1800 (i.e., 10 nanodiscs per ABCG2 dimer). After incubation of the mixture at 4 °C for 1 h, 0.8 g/mL of biobeads SM-2 were added and the mixture was rotated overnight at 4 °C to remove detergent and initiate nanodisc assembly. The following day, the biobeads were removed, and any remaining maltose was removed by three rounds of dilution and diafiltration against a 100 K MWCO filter. Excess nanodiscs were removed by rebinding the MBP-ABCG2 to amylose affinity resin and washing with 25 mM Tris (pH 8), 150 mM NaCl. The resin was resuspended in wash buffer and tobacco etch virus protease was added overnight to cleave MBP and release nanodisc incorporated ABCG2. The collected ABCG2 nanodiscs were concentrated, incubated with 2 mM ATP and 4 mM Mg$^{2+}$ for 45 min on ice, and finally injected over a Superose 6 gel filtration column in 25 mM Tris (pH 8), 150 mM NaCl. Peak fractions were pooled and concentrated to ~1 mg/mL for cryo-EM studies.

**EM sample preparation and data collection.** Prior to freezing grids for cryo-EM nanodisc reconstituted ABCG2 at a concentration of ~1 mg/mL was incubated with 75 μM MXN, SN38, or imatinib on ice for 45 min. In the case of apo ABCG2 the samples were not incubated with any compounds and applied directly to cryo-EM grids. A 3 μL volume of sample was applied to glow-discharged Quantifoil R1.2/1.3 holey carbon grids and blotted for 2.5 s on a Cryoplunge 3 system (Gatan) before being plunge frozen in liquid ethane cooled by liquid nitrogen. Cryo-EM images of apo, MXN, and SN38 bound ABCG2 were collected at liquid nitrogen temperature on a FEI F30 Polara equipped with a K2 Summit detector. Images collected on the Polara utilized a data collection strategy with a single shot per hole and a single hole per stage move. Cryo-EM images of ABCG2 with imatinib were collected on a Titan Krios equipped with a K3 detector. Images collected on the Titan Krios utilized a data collection strategy applying image shift and beam tilt to collect three shots per hole and four holes per stage move. Movies were recorded in super-resolution (Polara, K2) or counting mode (Krios, K3) with SerialEM data collection software[39]. The details of EM data collection parameters are listed in Extended Data Table 1.

**EM image processing.** EM data were processed as previously described with minor modifications[40]. Dose-fractionated super-resolution movies were binned over 2 × 2 pixels, and beam-induced motion was corrected using the program MotionCor2[41]. Defocus values were calculated using the program CTFFIND4[42]. Particle picking was performed using a semi-automated procedure implemented in Simplified Application Managing Utilities of EM Labs (SAMUEL)[43]. Two-dimensional (2D) classification of selected particle images was performed with "samclasscas.py", which uses SPIDER operations to run 10 cycles of correspondence analysis, K-means classification and multi-reference alignment, or by RELION 2D classification. Initial models for 3D classification were generated by projection matching of 2D class averages generated from samclasscas.py against a cylinder volume with approximate dimensions of ABCG2 particles, followed by projection matching refinement with SPIDER[43]. 3D classification and refinement were carried out in RELION2.0 or RELION3.0[44]. Unless otherwise indicated, all 3D classification and refinement steps were carried out with the application of C2 symmetry.

For the apo and imatinib datasets which display a single overall protein conformation, initial rounds of 3D classification were used to remove bad particles, followed by refinement of all particles in good classes into a single volume. A mask was then applied around the protein to avoid signal from the nanodisc, and further 3D classification with no alignment and a tau_fudge factor of 40 was performed to isolate particle populations with the propensity to reach higher resolutions. The orientation parameters of particles in the best class were then refined using the "auto-refine" procedure in RELION.

In the case of MXN and SN38 datasets where two predominant conformations were observed early in 3D classification, we utilized a 3D classification scheme with residual signal subtraction to focus on conformational differences in the transmembrane region. Masked 3D classification with residual signal subtraction was performed following a previously published procedure[45]. Masks were constructed to focus the classification on only the transmembrane helices, and omit signal from the nanodisc, NBDs, and extracellular loops. The orientation parameters of the homogenous set of particle images in selected 3D classes were iteratively refined to yield higher resolution maps using the "auto-refine" procedure in RELION.

All refinements followed the gold-standard procedure, in which two half datasets are refined independently. The overall resolutions were estimated based on the gold-standard Fourier shell correlation (FSC) = 0.143 criterion. Local resolution variations were estimated from the two half data maps using ResMap[46] or local resolution determination in RELION3.0. The amplitude information of the final maps was corrected by using "relion_postprocess" in RELION3.0 or the program bfactor.exe[47]. The number of particles in each dataset and other details related to data processing are summarized in Extended Data Table 1.

**Model building and refinement**. In order to model the apo-closed conformation, the high-resolution structure of human ABCG2 bound to the 5D3 Fab and MZ29 inhibitor (PDBid 6ETI) was used as an initial template for model building and refinement. The 5D3 Fab and all ligands were removed from the structure, leaving just the ABCG2 TM and NBD domains. TM5 was manually shifted and rotated to match the proper helical character and sequence register observed in the closed conformation map. Similarly, the unraveled portion of TM2 was manually adjusted in COOT to fit the apo-closed conformation map. The model was then iteratively adjusted manually in COOT and refined in real-space with phenix. real_space_refine.

The structure of human ABCG2 bound to the 5D3 Fab and MZ29 inhibitor (PDBid 6ETI) was also used as an initial template for model building and refinement for inward facing ABCG2 structures. The 5D3 Fab and all ligands were removed from the structure, leaving just the ABCG2 TM and NBD domains. The resulting model was rigid body fit in UCSF Chimera[48] into the inward facing conformation map of ABCG2 bound to imatinib, MXN, SN38 and then refined by iterative rounds of manual adjustment in COOT[49], with manual placement and real-space refinement of ligands, followed by real-space refinement in phenix. real_space_refine[50]. Chemical descriptions and refinement restraints for MXN and imatinib were obtained from the CCP4[51] monomer library and phenix.ready_set. A molecule of SN38 was manually built in JLigand[52], and refinement restraints were generated in phenix.ready_set. Final calculation of map vs model FSC was performed by first simulating a map from the refined atomic models to Nyquist frequency using UCSF Chimera, and then using this simulated map to calculate FSC with the final map from Relion autorefine.

**5D3 antibody shift assay**. For analysis of 5D3 antibody binding to ABCG2 in intact cells, a stable HEK293F cell line expressing wild-type human ABCG2 with an N-terminal GFP tag (GFP-ABCG2) was constructed. GFP-ABCG2 in the pcDNA3 vector was transfected into HEK293F cells with lipofectamine 2000 and maintained in Freestyle293 medium supplemented with 5% fetal bovine serum (FBS). Cells that had stably integrated GFP-ABCG2 were selected with geneticin and maintained in Freestyle293 medium with 5% FBS and 200–400 μg/mL geneticin. Expression of functional GFP-ABCG2 was verified by fluorescent size-exclusion chromatography, fluorescence microscopy, and flow cytometry analysis.

Stable GFP-ABCG2 cells were detached from culture plates with TrypLE dissociation reagent, washed with Freestyle293 medium, and resuspended in 100 μL of primary antibody labeling solution (phosphate-buffered saline (PBS) containing 0.5% (w/v) bovine serum albumin (BSA) and 1 μg/mL 5D3 antibody). Here appropriate MXN, SN38, imatinib, or Ko143 were included in the primary antibody labeling solution. Cells were incubated with primary antibody at 37 °C for 45 min with gentle agitation, washed with phosphate-buffered saline plus 0.5% BSA, and resuspended in 100 μL of secondary antibody labeling buffer (phosphate-buffered saline containing 0.5% (w/v) BSA and goat-anti-mouse IgG conjugated with PE). Here appropriate MXN, SN38, imatinib, and Ko143 were also included in the secondary antibody labeling solution. Cells were stained with secondary antibody for 30 min at 37 °C with gentle agitation before being washed with phosphate-buffered saline containing 0.5% (w/v) BSA and resuspended in the same buffer. Stained cells were maintained on ice and immediately used in flow cytometry analysis.

Stained cells were analyzed on a BD FacsCalibur flow cytometer with a 488 nm Argon blue laser. Cells initially gated on forward and side-scatter were first analyzed for GFP fluorescence in the FL1 channel with a 530/30 nm bandpass filter. A second gate around the predominant population with high GFP fluorescence was used to select cells with a high level of ABCG2 expression[20]. Gated cells were then analyzed for 5D3 antibody binding by monitoring PE staining in the FL2 channel with a 585/42 nm bandpass filter.

**ATPase activity measurements**. ATPase activity of nanodisc reconstituted ABCG2 was measured as previously described[53]. Briefly, 1 μg of nanodisc

reconstituted ABCG2 was incubated in 25 mM Tris, pH 8, 150 mM NaCl, 4 mM ATP, and 4 mM MgCl$_2$ for 30 min at 37 °C. Where appropriate transport substrates or inhibitors were included in the reaction mixture at the indicated concentrations. The reaction was stopped by the addition of 50 μL 12% (w/v) SDS. Then 100 μL of 6% (w/v) ascorbic acid and 1% (w/v) ammonium molybdate in 1 N HCl, followed by 150 μL of 25 mM sodium citrate, 2% (w/v) sodium metaarsenite, and 2% (v/v) acetic acid was added. Absorbance at 850 nm was measured using a SpectraMax M5 spectrophotometer (Molecular Devices). Potassium phosphate (KH$_2$PO$_4$) solutions were used to construct a standard curve to determine the total concentration of released phosphate. Data was plotted and analyzed in GraphPad Prism 8.

**Disulfide crosslinking**. For disulfide crosslinking analysis WT, Cys603Ser, Val534Cys, and Ala537Cys ABCG2 constructs with an N-terminal GFP were cloned into the pcDNA3 vector (mutagenesis primers listed in Supplementary Table 3). Site-directed mutants were constructed with the Q5 site-directed mutagenesis kit according to manufacturer's protocols. Constructs were transfected with 25 kDa linear polyethyleneimine into adherent HEK293F cells maintained in Freestlye293 media with 5% FBS. After allowing for protein expression for ~36 h, cells were incubated with the indicated transport inhibitors or substrates at a concentration of 20 μM (or 10 mM for orthovanadate) for 20 min. During the incubation period a solution of 6 mM (Cu$^{2+}$)/(1,10-phenanthroline)$_3$ was prepared by combining 500 mM CuSO$_4$ in water with 500 mM 1,10-phenanthroline in ethanol. Following the 20-min incubation period the media was replaced with media containing the same inhibitors or substrates and 300 μM (Cu$^{2+}$)/(1,10-phenanthroline)$_3$ and oxidation was allowed to proceed for 45 min. Cells were then washed with PBS, resuspended in 25 mM Tris pH 8, 150 mM NaCl, 1% DDM, 0.2% CHS, and protease inhibitors and incubated on a nutating rocker at 4 °C for 1 h to solubilize membranes. Samples were spun at ~15,000 × g and the soluble fraction was mixed with SDS-PAGE loading buffer containing 40 mM EDTA and 40 mM N-ethyl maleimide. Samples were subjected to non-reducing SDS-PAGE, and the resulting gels were visualized for in-gel GFP fluorescence using an Amersham 600 RGB imaging system.

**Thermal shift assay**. Stable N-GFP WT ABCG2 cells described above were resuspended in 25 mM Tris pH 8, 150 mM NaCl, 1% DDM, 0.2% CHS, and protease inhibitors and incubated on a nutating rocker at 4 °C for 1 h to solubilize membranes. Solubilized samples were incubated on ice with 20 μM of the indicated compound before being transferred to a thermocycler and incubated at the indicated temperature for 10 min. After the incubation the samples were immediately chilled on ice and passed through a 0.2 μM spin filter to remove aggregates. Samples were injected onto a Sepax SEC500 column attached to a Waters HPLC equipped with a fluorescence detector to monitor GFP fluorescence with excitation/emission wavelengths of 488/520 nm. Peak intensity as a function of incubation temperature was plotted in GraphPad Prism 8.

**Reporting summary**. Further information on research design is available in the Nature Research Reporting Summary linked to this article.

## Data availability

Data supporting the findings of this manuscript are available from the corresponding author upon reasonable request. A reporting summary for this Article is available as a Supplementary Information file.

The source data underlying Fig. 2c, f and Supplementary Fig. 1E are provided as a source data file.

The three-dimensional cryo-EM density maps of ABCG2 in nanodiscs have been deposited in the Electron Microscopy Data Bank under accession numbers EMD-21436 (apo), EMD-21437 (imatinib), EMD-21438 (MXN-inward), EMD-21439 (MXN-closed), EMD-21440 (SN38-inward), and EMDB-21441 (SN38-closed).

Atomic coordinates for the atomic models of ABCG2 have been deposited in the Protein Data Bank under accession numbers:
PDB 6VXF (apo) [https://doi.org/10.2210/PDB6VXF/pdb],
PDB 6VXH (imatinib) [https://doi.org/10.2210/pdb6VXH/pdb],
PDB 6VXI (MXN-inward) [https://doi.org/10.2210/pdb6VXI/pdb],
PDB 6VXJ (SN38-inward) [https://doi.org/10.2210/pdb6VXJ/pdb].

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

## Acknowledgements

The authors are grateful to Z. Li, S. Sterling, S. Rawson, and R. Walsh at the Harvard Cryo-Electron Microscopy Center for Structural Biology for EM technical support. The authors thank the members of the Liao group for helpful discussion and comments on the manuscript. B.J.O. was supported by a Postdoctoral Fellowship, PF-17-212-01-DMC, from the American Cancer Society.

## Author contributions

M.L. conceived the project. B.J.O. performed all experiments including molecular cloning, protein expression and purification, cryo-EM imaging and processing, model building and refinement, ATPase assays, chemical crosslinking, and flow cytometry. Both authors analyzed the data and wrote the manuscript.

## Competing interests

The authors declare no competing interests.
