## [Peer Review File · Nature Communications]

Reviewers' comments:

Reviewer #1 (Remarks to the Author):

The paper by Orlando and Liao presents some exciting new cryo-EM data on the drug binding site within the central cavity of the multidrug pump ABCG2.

Two different structures are presented; the first obtained in the absence of drug reveals a highly unexpected conformation particularly with respect to the transmembrane domains, in which TM helices 5 and 2. This novel TMD arrangement for ABCG2 was supported by biochemical experiments using conformation-sensitive antibody binding and disulphide cross-linking. The second class of structure are drug-bound structures: notably the combination of the approaches enables the authors to discriminate between true substrate substrates (mitoxantrone and SN38) and transporter inhibitors that bind at the same site (imatinib). The authors present a compelling model to encapsulate this, and I think this paper would be of great interest and relevance to the ABC transporter community.

Major:

The results do not give any indications of the number of independent experiments and technical repeats. This is a really surprising omission which is of the highest priority to rectify.

Lines 440-445. Can the authors provide some clarity regarding the final step of their purification? The protein is incubated with Mg.ATP prior to final HPLC analysis. For an "apo" structure this seems to be a very strange final step as it could introduce conformational heterogeneity into the preparation.

Lines 320-322 and Lines 448-450. The authors use concentrations of drugs in their EM that are so much higher than one would expect to see (75 microM). I would expect affinities of these drugs to be in the sub microM range. Some comment here is needed. This becomes relevant when considering their suggestion that drugs provide a bias in the conformation of ABCG2; have the authors tested if the relative proportions of the two predominant conformations (lines 254) are sensitive to the applied drug concentration? One would fully expect this to be the case. Or that the relative proportions of the two predominant conformations (lines 254) are sensitive to the duration of drug incubation?

Lines 154-171, 536-552; Figure 2. The cross-linking data described in this section is evidence for the sampling of the unusual conformation in figure 1. I do wonder why the work was performed in cells rather than membranes – one might expect that in membranes the cross-linking in the presence of drugs was much more substantial as there would be no drug export, which I suspect is why the percentage cross-linking is low. Were time courses performed for the cross-linking experiments? This might be relevant as kinetic information might be useful for deriving relative apparent drug affinities.

Lines 185-215. I really liked the flow cytometry data provided which is genuinely elegant, but the ATPase assay data is less so. The expectation for an inhibitor is that ATPase activity would be decreased in a dose-dependent manner, and this is observed for Ko143 and for imatinib. For the two substrates (SN38 and mitoxantrone) I think the ATPase assay data are ambiguous. I would expect dose-dependent stimulation not inhibition. The authors assert that "this was consistent with that obtained for E1S" (estrone-1-sulphate; reference quoted 24) but comparison of the current paper Figure S1D with the figure S2 in the cited reference 24 (Manolaridis et al., 2018) does not support this. E1S shows a clear dose-dependent stimulation of ATPase activity from which an EC50 concentration could be derived. The current ATPase data does not fully support the authors narrative here, although it does provide evidence that imatinib is an inhibitor.

Lines 204-220 with no values for "n" and no evidence of experimental variability the data in Figure

S1E should be removed and the test should be modified as well as there is no evidence of thermostabilisation as presented.

Minor:

From a narrative point of view I think the authors need to be clear about previous structures and their own structure throughout the paper. For example on line 107 the phrase "in the closed conformation" should read "in our apo-closed conformation" so that the reader knows exactly which structure is referred to. This is true elsewhere in the paper with respect to both their own structure and previous Locher group structures (e.g. lines 151-152). Please can the authors use the same terms for any references to these previous structures. I would prefer to see the pdb codes given in parenthesis to help the reader here. Figure legends would benefit from the same degree of clarity.

Line 111: I cannot see the residue Met541 in Figure 1D – can the authors check this.

Line 147-15: can the authors provide a supplementary table listing the predicted distances between the two residue pairs across their apo-closed structure, the Locher inward open structure and the Locher ATP-bound structure. This would really help the reader see that the predictions about residues being "too distant" are valid

Lines 340-350. In the discussion the authors attempt explanation of their data with respect to the well described R482 residue but there are also publications describing mutation to M549, F545 and M523 that may be relevant here.

Reviewer #2 (Remarks to the Author):

Orlando et al., report the cryo-EM structures of the human ABC transporter ABCG2 in apo and three different anti-cancer drugs bound states (imatinib, mitoxantrone, and SN38). While the structures of human ABCG2 in fab, inhibitors and ATP bound states had been described previously (Taylor et. al 2017, Jackson et. al 2018, Manolaridis et. al 2018), the new ABCG2 apo structure described here reveals a novel closed conformation. Compared to previous ABCG2 structures in the inward-open state, both of TM2 and TM5 of ABCG2 in closed state exhibit significantly distinct conformations. Using site-directed disulfide crosslinking, Orlando et al. experimentally show that this closed conformation of ABCG2 exists in vivo and thus represents a functional relevant state. The structures of ABCG2 bound with three different anti-cancer drugs determined in this work adopt similar inward-open conformations as those previously observed in the structures of ABCG2 with inhibitors bound. Based on the biochemical and structural studies, the authors propose that imatinib is more effective than another two anti-cancer drugs in stabilizing the inward-open conformation of ABCG2.

Overall, the cryo-EM and biochemical works looks solid, and are of high technical quality. By elucidating the structure of ABCG2 in a novel closed conformation, this manuscript substantially advances our knowledge and understanding of the transporting mechanism of ABCG2, and provide structural implications for other members in ABC transporter family. The structures of ABCG2 with three different anti-cancer drugs bound help understanding how these drugs are transported out from the cancerous cells, which will facilitate the design of better drugs for chemotherapy. Considering these, this work can potentially make big impact on the field of ABC transporter and chemotherapy.

I have only minor comments:

(1) The binding modes between ABCG2 and either of three anti-cancer drugs are similar. Only one drug molecular is bound approximately at the 2-fold symmetrical axis of ABC2G dimer. Therefore, the overall structure ABCG2 with one drug bound is actually related by a pseudo C2 symmetry. The particle misalignment caused by the pseudo symmetry will definitely affect the structural

determination of the drug bound ABCG2 structure. Under this circumstance, the accuracy of modelling a small compound might be poor. My sense is that the overall fitting of these compounds is relatively accurate, but the authors should briefly discuss the degree of uncertainty in modeling the drug, owing to the pseudo symmetry, in the main text.

(2) The author claim that the imatinib is more effective than another two drugs in the stabilization of the inward-open conformation of ABCG2. However, the cryo-EM map of ABCG2 with imatinib bound was solved at relatively lower resolution, as compared with other cryo-EM maps. Could authors provide an explanation? I notice that the data collection strategy for ABCG2/imatinib sample is different from those used in other samples. Could this be the reason?

(3) The workflows of particle classification for 4 different dataset are quite different. The authors should elaborate the procedures of imaging processing for each dataset in the method section.

(4) The author should describe how the 3D initial model was generated.

(5) The authors should show the representative 2D class averages at least for one of the 4 datasets.

(6) The author should describe how the FSC between model and map was calculated.

(7) The title of Supplementary Figure 7: "Cryo-EM Data Analysis of ABCG2-mitoxantrone" should be changed to "Cryo-EM Data Analysis of ABCG2-MXN" to be consistent with the title of Supplementary Figure 6.

Response to referees

We thank the reviewers for their time and constructive comments, which have helped to strengthen our manuscript considerably. Below we include the initial reviewer comments in italicized and indented text, followed by our responses.

Reviewer #1

The paper by Orlando and Liao presents some exciting new cryo-EM data on the drug binding site within the central cavity of the multidrug pump ABCG2.

Two different structures are presented; the first obtained in the absence of drug reveals a highly unexpected conformation particularly with respect to the transmembrane domains, in which TM helices 5 and 2. This novel TMD arrangement for ABCG2 was supported by biochemical experiments using conformation-sensitive antibody binding and disulphide cross-linking. The second class of structure are drug-bound structures: notably the combination of the approaches enables the authors to discriminate between true substrate substrates (mitoxantrone and SN38) and transporter inhibitors that bind at the same site (imatinib). The authors present a compelling model to encapsulate this, and I think this paper would be of great interest and relevance to the ABC transporter community.

We thank the reviewer for his/her favorable comments.

The results do not give any indications of the number of independent experiments and technical repeats. This is a really surprising omission which is of the highest priority to rectify.

Where appropriate in figure legends and methods sections we have now included information for the number of independent experiments conducted and statistical analyses performed.

Lines 440-445. Can the authors provide some clarity regarding the final step of their purification? The protein is incubated with Mg-ATP prior to final HPLC analysis. For an “apo” structure this seems to be a very strange final step as it could introduce conformational heterogeneity into the preparation.

Initial attempts to obtain cryo-EM structures of ABCG2 in the absence of conformation specific antibody fragments or high-affinity inhibitors produced 3D reconstructions with very heterogeneous transmembrane helices and nucleotide binding domains. In order to overcome this heterogeneity, we attempted several methods of biochemical optimization. Early in the optimization process we subjected the same ABCG2-nanodisc preparation to incubation in different conditions such as continuous ATP hydrolysis (“worked”), or with nucleotide analogs such as AMP-PNP or ADP-vanadate. Samples subjected to these conditions were then imaged by negative-stain electron microscopy. Surprisingly, and somewhat counter-intuitively, we observed that ABCG2-nanodiscs that had been “worked” by supplying ATP-Mg for a continuous hydrolysis reaction appeared more conformationally homogeneous (at a low resolution negative-stain level) than samples incubated in conditions that promote nucleotide binding but not hydrolysis. These results suggested that ABCG2 that has been “worked” by undergoing continuous rounds of ATP binding and hydrolysis becomes more conformationally homogeneous.

Similar results demonstrating improved stability and activity following prolonged ATP binding and hydrolysis have also been reported by Livnat-Levanon et al. for the bacterial ABC importer BtuC₂D₂ (reference below).

Livnat-Levanon, N., A. I. G., Ben-Tal, N. & Lewinson, O. 2016. The uncoupled ATPase activity of the ABC transporter BtuC2D2 leads to a hysteretic conformational change, conformational memory, and improved activity. *Sci Rep*, 6, 21696.

It is interesting to note that BtuC₂D₂ and ABCG2 are similar in that no domain swapping between the two halves of the protein is observed. Based on the increased conformational homogeneity observed by negative stain for ABCG2 that had been “worked” through ATP hydrolysis, along with previous reports of similar stabilization for another ABC transporter system, we incorporated a step in our purification scheme to allow ABCG2 to cycle through several rounds of ATP hydrolysis before the final gel filtration step. Indeed, incorporation of this ATPase cycling step prior to final gel-filtration lead to markedly improved conformational homogeneity in cryo-EM 3D classification, mirroring the results seen at low resolution with negative stain. It is important to note that ATP-Mg was not included in the final gel-filtration buffer and should be removed from the sample by this final purification step prior to cryo-EM grid vitrification. The lack of any observable nucleotide binding domain dimerization or nucleotide density in our cryo-EM maps provides further evidence that ATP-Mg was efficiently removed in the final gel-filtration column.

Lines 320-322 and Lines 448-450. The authors use concentrations of drugs in their EM that are so much higher than one would expect to see (75 microM). I would expect affinities of these drugs to be in the sub microM range. Some comment here is needed. This becomes relevant when considering their suggestion that drugs provide a bias in the conformation of ABCG2; have the authors tested if the relative proportions of the two predominant conformations (lines 254) are sensitive to the applied drug concentration? One would fully expect this to be the case. Or that the relative proportions of the two predominant conformations (lines 254) are sensitive to the duration of drug incubation?

Cryo-EM grids of ABCG2 were frozen at a protein concentration of ~1.2mg/mL (monomer) which equates to a molar concentration of ~15 μ M. We chose to incubate ABCG2 with 75 μ M drug to provide a 5-fold molar excess of compound over potential binding sites, thus ensuring maximal binding and potential for visualization of the drug molecules in the resultant cryo-EM maps. It is important to note that inhibitory and conformational effects of imatinib were seen at much lower concentrations (1-10 μ M) in ATPase and antibody shift assays.

At this time we have not extensively evaluated the effect that drug concentration or incubation time has on the proportion of the two observed conformations. Such an analysis with cryo-EM would require **several large datasets** at different drug concentrations and incubation times with each individual compound, which is beyond the scope of the current manuscript. Furthermore, in our opinion cryo-EM is likely not the ideal assay for this type of analysis given the fact that relatively high drug concentrations are required for structure-based approaches, and that 3D classification will be biased towards more conformationally homogeneous particle populations. We are currently developing biochemical and spectroscopic assays to further probe the effect of different drug concentrations and incubation periods on ABCG2 conformation, which will allow for analysis at more physiological concentrations.

Lines 154-171, 536-552; Figure 2. The cross-linking data described in this section is evidence for the sampling of the unusual conformation in figure 1. I do wonder why the work was performed in cells rather than membranes – one might expect that in membranes the cross-linking in the presence of drugs was much more substantial as there would be no drug export, which I suspect is why the percentage cross-linking is low.

Our initial crosslinking experiments were all performed in crude membrane vesicles derived from transiently transfected HEK293F cells. While we indeed observed more robust cross-linking in membrane vesicle preparations compared to intact cells, we also found that cross-linking in vesicles was relatively insensitive to added compounds, including Ko143, imatinib, and vanadate. The precise reason for this drug insensitivity

remains unknown. It is possible that more elaborate methods beyond crude membrane preparation (such as density gradient ultracentrifugation) to isolate more homogeneous membrane vesicles could provide more consistent results. Nevertheless, we found crosslinking in intact cells to be repeatable and accurately report on the conformation expected when used with controls (Ko143 and vanadate). Crosslinking in intact cells also provides a more physiologically relevant environment compared to crude membranes.

Were time courses performed for the cross-linking experiments? This might be relevant as kinetic information might be useful for deriving relative apparent drug affinities.

We performed crude initial time courses for cross-linking to establish optimal conditions for the assay. However, extensive time courses in the presence of each individual drug were not performed. In our cell-based cross-linking assay several factors could influence the kinetics of cross-linking, such as the rate of drug penetration into and/or across the cell membrane and binding to non-target sites. For these reasons we chose an end-point readout for the cross-linking assay simply to monitor the overall effect that each compound induces on the conformation of ABCG2. While we agree with the reviewer that relative apparent drug affinities would be very useful information, such values may not be readily derived from our cross-linking assay in cells. Future studies using more targeted assays such as microscale thermophoresis, radioligand binding, or fluorescence polarization would likely provide much more accurate and reliable apparent drug affinities.

Lines 185-215. I really liked the flow cytometry data provided which is genuinely elegant, but the ATPase assay data is less so. The expectation for an inhibitor is that ATPase activity would be decreased in a dose-dependent manner, and this is observed for Ko143 and for imatinib. For the two substrates (SN38 and mitoxantrone) I think the ATPase assay data are ambiguous. I would expect dose-dependent stimulation not inhibition. The authors assert that “this was consistent with that obtained for E₁S” (estrone-1-sulphate; reference quoted 24) but comparison of the current paper Figure S1D with the figure S2 in the cited reference 24 (Manolaridis et al., 2018) does not support this. E₁S shows a clear dose-dependent stimulation of ATPase activity from which an EC₅₀ concentration could be derived. The current ATPase data does not fully support the authors narrative here, although it does provide evidence that imatinib is an inhibitor.

We thank the reviewer for his/her careful analysis of the flow cytometry and ATPase data. Indeed, while the ATPase data clearly demonstrate that imatinib suppresses the ATPase activity of ABCG2 similar to the bona fide inhibitor Ko143, MXN and SN38 have almost no effect on the rate of ATP hydrolysis. In our original manuscript we cited Manolaridis et al. (2018) to suggest that E₁S also does not show stimulation of ATPase activity. The reviewer is correct that all assays in Manolaridis et al. (2018) demonstrate dose-dependent stimulation of ATPase activity by E₁S. However, all of the assays in that reference were performed in proteoliposomes. We should have also referenced Taylor et al. (2017) in which the stimulation of ABCG2 ATPase activity by E₁S was tested both in proteoliposomes and lipid nanodiscs. Whereas Taylor et al. (2017) observed stimulation by E₁S in proteoliposomes, no stimulation was observed in nanodiscs, similar to what we observed with MXN and SN38. We thank the reviewer for spotting this error. We have added the appropriate reference to Taylor et al. (2017) in our manuscript.

While we agree that substrates such as MXN and SN38 would be expected to stimulate ATPase activity, thus far no stimulation of ATPase activity has been observed for nanodisc incorporated ABCG2 with any transported substrate tested by us or other groups. It is important to note that in Taylor et al. (2017) the specific ATPase activity of nanodisc incorporated ABCG2 is higher than that of ABCG2 in proteoliposomes. Indeed, the ATPase activity of nanodisc incorporated ABCG2 was found to be essentially identical to that seen with substrate stimulation of liposome incorporated transporter. The aim of our ATPase data was to demonstrate that similarly

to what was observed in cryo-EM, flow cytometry, and cross-linking assays, the substrates MXN and SN38 display clearly different effects than imatinib on ABCG2 ATPase activity.

Lines 204-220 with no values for “n” and no evidence of experimental variability the data in Figure S1E should be removed and the test should be modified as well as there is no evidence of thermostabilisation as presented.

We have repeated the experiment a total of three independent times (n=3) and included relevant statistical analysis in the figure legend. Although the thermostabilization afforded by imatinib binding is small compared to Ko143, the effect is statistically significant.

From a narrative point of view I think the authors need to be clear about previous structures and their own structure throughout the paper. For example on line 107 the phrase “in the closed conformation” should read “in our apo-closed conformation” so that the reader knows exactly which structure is referred to. This is true elsewhere in the paper with respect to both their own structure and previous Locher group structures (e.g. lines 151-152). Please can the authors use the same terms for any references to these previous structures. I would prefer to see the pdb codes given in parenthesis to help the reader here. Figure legends would benefit from the same degree of clarity.

We have revised the manuscript to more clearly reflect exactly which structures are being referred to, and included PDB codes as suggested by the reviewer. We believe this has significantly improved the clarity of the manuscript, and we thank the reviewer for his/her constructive comments.

Line 111: I cannot see the residue Met541 in Figure 1D – can the authors check this.

Met541 is present and labeled in Figure 1D. The label was absent from Figure 1C. We added this label as well as several others to Figure 1C for clarity.

Line 147-15: can the authors provide a supplementary table listing the predicted distances between the two residue pairs across their apo-closed structure, the Locher inward open structure and the Locher ATP-bound structure. This would really help the reader see that the predictions about residues being “too distant” are valid.

We have added the table to the supplement.

Lines 340-350. In the discussion the authors attempt explanation of their data with respect to the well described R482 residue but there are also publications describing mutation to M549, F545 and M523 that may be relevant here.

We thank the reviewer for pointing out previous studies with mutational analysis that may be relevant to our manuscript. However, the reason we specifically highlighted the well-known R482 residue is because mutations at this location completely abolish the conformation and ATPase specific effects of imatinib. At this time the specific effects of other mutations on ABCG2 structure and function remain unclear. Whereas we detected no effect of a M549A mutation on MXN transport in a flow cytometry-based efflux assay (data not shown), and Manolaridis et. al. (2018) report no effect of the same mutation on ATPase activity or E₁S transport, Haider et. al. (2015) present data suggesting that the M549A mutation has significant impacts on MXN transport.

Haider, A.J., et. al. *Bioscience Reports* (2015) **35**, e00241, doi:10.1042/BSR20150150

Cox et. al. (2018) perform a flow cytometry efflux assay to analyze the effect of M523A or F545A mutants on MXN transport. Although M523A and F545A showed slightly reduced MXN transport, these effects are minor (not statistically significant for M523A), and the F545A mutant shows significantly reduced overall expression.

Cox, M.H. et. al. *Biochemical Journal* (2018) 475 1553–1567

We appreciate the reviewer's suggestion on publications of additional ABCG2 mutants. However, for the reasons mentioned above we prefer to refrain from including these mutants in our current discussion. Significantly more biochemical work is needed to tease out all of the nuances of different mutational effects with different ABCG2 substrates/inhibitors.

Reviewer #2

Orlando et al., report the cryo-EM structures of the human ABC transporter ABCG2 in apo and three different anti-cancer drugs bound states (imatinib, mitoxantrone, and SN38). While the structures of human ABCG2 in fab, inhibitors and ATP bound states had been described previously (Taylor et. al 2017, Jackson et. al 2018, Manolaridis et. al 2018), the new ABCG2 apo structure described here reveals a novel closed conformation. Compared to previous ABCG2 structures in the inward-open state, both of TM2 and TM5 of ABCG2 in closed state exhibit significantly distinct conformations. Using site-directed disulfide crosslinking, Orlando et al. experimentally show that this closed conformation of ABCG2 exists in vivo and thus represents a functional relevant state. The structures of ABCG2 bound with three different anti-cancer drugs determined in this work adopt similar inward-open conformations as those previously observed in the structures of ABCG2 with inhibitors bound. Based on the biochemical and structural studies, the authors propose that imatinib is more effective than another two anti-cancer drugs in stabilizing the inward-open conformation of ABCG2.

Overall, the cryo-EM and biochemical works looks solid, and are of high technical quality. By elucidating the structure of ABCG2 in a novel closed conformation, this manuscript substantially advances our knowledge and understanding of the transporting mechanism of ABCG2, and provide structural implications for other members in ABC transporter family. The structures of ABCG2 with three different anti-cancer drugs bound help understanding how these drugs are transported out from the cancerous cells, which will facilitate the design of better drugs for chemotherapy. Considering these, this work can potentially make big impact on the field of ABC transporter and chemotherapy.

We thank the reviewer for their favorable comments.

The binding modes between ABCG2 and either of three anti-cancer drugs are similar. Only one drug molecular is bound approximately at the 2-fold symmetrical axis of ABC2G dimer. Therefore, the overall structure ABCG2 with one drug bound is actually related by a pseudo C2 symmetry. The particle misalignment caused by the pseudo symmetry will definitely affect the structural determination of the drug bound ABCG2 structure. Under this circumstance, the accuracy of modelling a small compound might be poor. My sense is that the overall fitting of these compounds is relatively accurate, but the authors should briefly discuss the degree of uncertainty in modeling the drug, owing to the pseudo symmetry, in the main text.

The reviewer is correct that the drug binding location lies directly along the 2-fold axis between ABCG2 monomers, which complicates fitting of the drug molecules into the resultant cryo-EM maps. Several attempts were made to break this pseudo-symmetry problem through particle symmetry expansion and/or classification without alignment. However, these attempts were largely unsuccessful. The pseudosymmetry problem is more pronounced in the maps with MXN and SN38 as compared to imatinib. In order to clarify this issue for the reader we have included the following in the main text of the manuscript at line 270-276 ...

“An inherent complication in resolving drug-bound structures of ABCG2 with cryo-EM lies in the fact that the drug binding pocket lies directly along the 2-fold axis relating monomers of ABCG2. As a result, the particles in 3D reconstruction contain inherent 2-fold pseudo-symmetry, which causes the density for drug molecules to suffer from symmetry artifacts. Several attempts to break this pseudo-symmetry through symmetry expansion and/or classification without alignment were unsuccessful. Nevertheless, the drug densities for MXN, SN38, and imatinib are of sufficient quality to determine the overall binding orientation of each molecule..”

The author claim that the imatinib is more effective than another two drugs in the stabilization of the inward-open conformation of ABCG2. However, the cryo-EM map of ABCG2 with imatinib bound was solved at relatively lower resolution, as compared with other cryo-EM maps. Could authors provide an explanation? I notice that the data collection strategy for ABCG2/imatinib sample is different from those used in other samples. Could this be the reason?

Although the imatinib bound structure is determined to a slightly lower resolution than other structures, we would like to point out that this resolution difference is rather small. Indeed the interpretability of the imatinib map, particularly in the transmembrane region is similar to other maps in the manuscript. Several potential variables could lead to slightly lower resolution for the imatinib structure including relative ice thickness on cryo-EM grids, microscope alignments and stability, variability between sample preparations, etc. The largest variable that separates the imatinib dataset from others lies in the fact that this sample was collected on a different microscope, with a different detector and pixel size, and with a data collection strategy that incorporated beam-tilt to increase the image acquisition rate. Although beam-tilt correction in Relion 3.0 was used in the final processing steps of the imatinib dataset, the full potential of beam-tilt refinement can only be realized with accurate tracking of beam-tilt during data collection, and proper image grouping procedures during data processing. Unfortunately, tracking of beam-tilt with each image was not performed during the data collection for the imatinib structure. It is therefore likely that some degree of beam-tilt remains in the imatinib dataset, which in turn could limit the achievable resolution.

The workflows of particle classification for 4 different dataset are quite different. The authors should elaborate the procedures of imaging processing for each dataset in the method section.

We have updated the methods section to include more details of image processing for each dataset (line 474-485). The only difference in data processing between datasets involves the use of signal subtraction classification to focus on the transmembrane region in MXN and SN38 datasets where two predominant conformations of the transmembrane helices were clearly visible from earlier rounds of 3D classification.

The author should describe how the 3D initial model was generated.

We have added more details in the methods section (line 468-471) for initial 3D model generation. Briefly, the initial models were generated by first creating a cylinder volume with SPIDER and then using 2D class averages generated by samclasscas.py to perform projection matching refinement in SPIDER to reconstruct the initial volume. The initial model generated in this fashion was low pass filtered before use in 3D classification.

The authors should show the representative 2D class averages at least for one of the 4 datasets.

We have included 2D averages for the “apo-closed” dataset in Supplemental Figure 3.

The author should describe how the FSC between model and map was calculated.

We have added more details in the methods section (line 512-515) describing the calculation of FSC between model and map. Briefly, a simulated map at Nyquist frequency was calculated from the refined atomic coordinates in UCSF Chimera. This simulated model map was used to calculate FSC against the final map from Relion 3.0 autorefine.

The title of Supplementary Figure 7: “Cryo-EM Data Analysis of ABCG2-mitoxantrone” should be changed to “Cryo-EM Data Analysis of ABCG2-MXN” to be consistent with the title of Supplementary Figure 6.

We thank the reviewer for catching this error. The title has been changed as suggested.

REVIEWERS' COMMENTS:

Reviewer #1 (Remarks to the Author):

The authors have painstakingly addressed every point I raised and with the additional clarity this brings to the manuscript I believe this now better presents their outstanding research.

Reviewer #2 (Remarks to the Author):

The authors have satisfactorily addressed all my concerns. The manuscript has been improved substantially. I, therefore, strongly support the publication of this manuscript.